

**Chemical characterization of organic compounds involved in iodine-initiated new particle**
**formation from coastal macro-algal emission**
Yibei Wan[1], Xiangpeng Huang[2], Chong Xing[1], Qiongqiong Wang[1], Xinlei Ge[2], Huan Yu[1,*]
[1] School of Environmental Studies, China University of Geosciences, Wuhan, 430074, China
[2] Jiangsu Key Laboratory of Atmospheric Environment Monitoring and Pollution Control,
Collaborative Innovation Center of Atmospheric Environment and Equipment Technology, School
of Environmental Science and Engineering, Nanjing University of Information Science and
Technology, Nanjing 210044, China
* To whom correspondence should be addressed: yuhuan@cug.edu.cn
**Abstract**
Iodine-initiated new particle formation (I-NPF) has long been recognized in coastal hotspot
regions. However, no prior work has studied the exact chemical composition of organic compounds
and their role in the coastal I-NPF. Here we present an important complementary study to the
ongoing laboratory and field researches of iodine nucleation in coastal atmosphere. Oxidation and
NPF experiments with vapor emissions from real-world coastal macroalgae were simulated in a bag
reactor. On the basis of comprehensive mass spectrometry measurements, we reported for the first
time a variety of volatile precursors and their oxidation products in gas and particle phases in such a
highly complex system. Organic compounds overwhelmingly dominated over iodine in the new
particle growth initiated by iodine species. The identity and transformation mechanisms of organic
compounds were proposed in this study to provide a more complete story of coastal NPF from
low-tide macroalgal emission.
**1. Introduction**
Coastal new particle formation (NPF) may be driven by daytime low-tide emission of iodine
species from macroalgae fully or partially exposed to the air. The phenomenon was reported in
hotspot locations of west Europe, Australia and polar regions (O'dowd et al., 2002; Heard et al., 2006;
Mcfiggans et al., 2010; Whitehead et al., 2009; Sipilä et al., 2016; Allan et al., 2015; Baccarini et al.,
2020; Beck et al., 2021). In the southeast coastline of China, we reported intense iodine-initiated
NPF based on particle number size distribution and iodine measurements (Yu et al., 2019).



To simulate iodine-initiated NPF (I-NPF) in controlled laboratory conditions, $I_2$ or $CH_2I_2$ vapor
was usually photolyzed in the presence of ozone to provide nucleation precursors (Burkholder et al.,
2004; Jimenez et al., 2003; Monahan et al., 2012; Saunders and Plane, 2005; O'dowd et al., 2004;
Martín et al., 2020; He et al., 2021; Huang et al., 2022). Ashu-Ayem et al. (2012); Monahan et al.
(2012); Mcfiggans et al. (2004); Sellegri et al. (2005) and Sellegri et al. (2016) also investigated the
NPF from the vapors emitted by real-world macroalgal specimens or seawater in laboratory chamber
or apparatus. However, the focus of all above studies are emission rate, oxidation mechanisms or
nucleation pathways of iodine species. For example, positive correlations between particle
concentrations and $I_2$ or $CH_2I_2$ mixing ratios were usually observed (Burkholder et al., 2004; Jimenez
et al., 2003; Sellegri et al., 2005; Monahan et al., 2012). Kinetic studies in flow tube or CERN
CLOUD chamber proposed the clustering of iodine oxides ($I_xO_y$) or iodine oxoacids ($HIO_3$, $HIO_2$) as
nucleation mechanisms on the basis of photoionization TOF-MS (Martín et al., 2020), Api-TOF and
nitrate-Chemical Ionization Mass Spectrometer (CIMS) measurements (He et al., 2021). A recent
chamber study showed heterogeneous reaction between iodine oxide nanoparticle, meso-erythritol
(or glyoxal) and dimethylamine accelerated nanoparticle growth (Huang et al., 2022).
Until now, no prior work has investigated the exact chemical identity of organic compounds
(other than iodinated methane) and their role in I-NPF. The role of biogenic terpenes and
anthropogenic aromatics in continental NPF has been recognized for a long time (Donahue et al.,
2013). Their ozonolysis or photochemistry products have been investigated in depth by using
Electrospray Ionization Mass Spectrometry (ESI-MS) and more recently, CIMS (Nguyen et al., 2010;
Kundu et al., 2017; Kundu et al., 2012; Faxon et al., 2018; Wang et al., 2020; Riva et al., 2017; Yan
et al., 2020; Ehn et al., 2014). It is very likely that certain volatile organic compounds (VOCs)
emitted mutually with iodine or iodinated methane from coastal biota or biologically active sea
surface may also be involved in coastal I-NPF process and promote the growth of iodine particles.
To test this hypothesis, we conducted oxidation and NPF experiments with vapor emissions from
real-world coastal macroalgae in a bag reactor. A suite of mass spectrometric methods including
Inductively Coupled Plasma-MS (ICP-MS), Gas Chromatography-MS (GC-MS), iodide-CIMS and
ESI-orbitrap MS were applied to measure vapor precursors, gaseous products and particulate
products during the NPF process. Mass concentrations of total organic carbon (TOC) and total iodine
(TI) of new particles were compared to evaluate the relative importance of organics and iodine in
new particle growth. The identity and transformation mechanisms of organic compounds were
identified to provide a more complete story of coastal NPF from low-tide macroalgal emission. Our
study is thus complementary to prior laboratory and field studies of I-NPF, but has an emphasis on



organics.
**2. Experiments**
**2.1 Experimental apparatus and sample collection**
Similar to Potential Aerosol Mass (PAM) Oxidation Flow Reactor, a bag reactor was designed to
provide an oxidizing environment for simulating atmospheric oxidation processes of algae-emitted
VOCs. The bag reactor was made from 75 μm-thick fluorinated ethylene propylene (FEP) Teflon
(1.2 m×1.5 m, flat dimension). The volume of the bag at full inflation was determined
experimentally to be about 200 L. The bag was suspended vertically (Figure 1) and kept in the dark
or directly exposed to room light of fluorescent lamp. Because the purpose of this study is to
qualitatively measure the oxidation products of algae-emitted VOCs, wall loss, production rate and
other kinetic factors in the bag reactor were not evaluated. Fresh macroalgae (*Undaria pinnatifida*)
was collected from intertidal zone at Xiangshan gulf of east China coast and stored at -10 °C until the
experiments. 2 kg macroalgae was put in a 20 L Pyrex glass bottle that was filled with ~1 L natural
seawater. The specimens was partially exposed to the air to simulate tidal emersion of macroalgae. A
flow of particle-free ultra high purity (UHP) air blew algae-emitted VOCs out of the bottle and
merged with a diluting air flow before entering the bag reactor.
Two types of experiments were conducted. In the three ozonolysis experiments, ozone ($O_3$) was
generated by flowing an UHP air flow through a 5 Watts 185 nm UV lamp. The $O_3$ flow was fed just
before the bag reactor was fully inflated. Final $O_3$ concentration in the bag reactor was measured to
be ~200 ppbv using an ozone analyzer (Model 49i, Thermo-Fisher Scientific Inc.). In an additional
OH-enhanced experiment, the $O_3$/VOC mixture flow was directed through a 254 nm UV light before
entering the bag reactor. OH radicals were produced via the reaction $O_3+h\nu \rightarrow O_2+O(^1D)$ and
$O(^1D) +H_2O \rightarrow 2OH$.
Before each experiment, the bag was purged for several hours to reduce background particle
concentrations to below 1 $cm^{-3}$. The bag reactor was first operated in a static mode to monitor the
time evolution of gaseous products and particle size, and then in a dynamic mode to collect enough
particles for offline chemical analysis. In the static mode, the bag was first filled to full inflation with
the VOCs/$O_3$ flows. The flows were then shut down; a Scanning Mobility Particle Sizer (SMPS,
model 3936, TSI Inc., Shoreview, MN, USA) and an Aerodyne iodide-CIMS pulled two flows of 0.3
liters per minute (lpm) and 1.8 lpm out of the bag, respectively. The SMPS measured the particle
number size distribution from 14 to 600 nm.



In the dynamic mode, the VOCs/$O_3$ flow of 3 lpm was fed to the bag continuously, while the
SMPS and a vacuum pump (GAST Group Ltd.) pulled sample flows of 0.3 and 2.7 lpm, respectively,
out of the bag reactor. This resulted in an overall residential time of 67 minutes for the $O_3$/VOC
mixture in the fully inflated bag. The particles in the 2.7 lpm sample flow were collected onto a
Zefluor® PTFE membrane filter mounted in a filter inlet for gases and aerosols (FIGAERO) for
iodide-CIMS analysis, or alternatively, onto 47 mm diameter double quartz fiber filter pack mounted
in a filter holder for ESI-orbitrap MS, ICP-MS and TOC analysis. The front filter of the double filter
pack collected the particles, while the back filter placed downstream of the front filter was supposed
to adsorb the same amount of volatile species as the front filter.
**2.2 Chemical analysis**
Before the ozonolysis experiments, the algae-emitted VOCs in the bag reactor was collected by a
6-liter pre-evacuated stainless-steel canisters (Entech Instruments, Inc., Simi Valley, CA, USA) and
was analyzed using a quadrupole GC-MS system (model TH-300B, Wuhan Tianhong Instruments Co.
Ltd., Wuhan, China). The algae-emitted VOCs, as well as their gaseous and particulate products,
were also measured by the FIGAERO-iodide-CIMS. Iodide-adduct chemical ionization is well suited
for measuring oxygenated or acidic compounds with minimal fragmentation. More details of the
GC-MS and FIGAERO-iodide-CIMS measurements can be found in Supporting Material. The
theory and design of the two instruments were described by Wang et al. (2014) and Lopez-Hilfiker et
al. (2014).
The particles collected on quartz fiber filters were sent for offline quantification of TOC and TI,
as well as non-target analysis of organic compounds using ESI-orbitrap MS. The front and back
filters were treated, separately, following the procedure as below: the filter was ultrasonicated twice
with 10-mL water and acetone nitrile solvent mixture (v:v=1:1). The extract was filtered by a 0.2 μm
PTFE syringe filter and evaporated in a rotary evaporator to 0.5 mL. After being centrifuged for 30
min at 12000 rpm, the supernatant was collected for TI analysis by Agilent 1100 HPLC-7900
ICP-MS (Agilent Technologies, Santa Clara, CA, USA) and TOC analysis by a TOC analyzer
(Model TOC-5000A, Shimadzu, Japan). TI or TOC in the particles was obtained by subtracting the
amount on the back filter from that on the front filter. Nontarget analysis of organic compounds in
the supernatant was conducted using a Q Exactive hybrid Quadrupole-Orbitrap mass spectrometer
(Thermo Scientific, Bremen, Germany). The supernatant was directly infused by a syringe pump and
ionized in negative ESI source. All the ions in the m/z range from 50 to 500 Th were scanned with a
mass resolution of 70000. The chemically sound CHO molecular formulas were computed with a



mass tolerance of ±2 ppm for these ions. Only the compounds that existed solely in the front filter or
with ion intensity in the front filter higher than that in the back filter by a factor of 3 were regarded
as the organic compounds in the particle phase.
**3. Results and discussion**
**3.1 Relative mass contribution of organic carbon and iodine to new particles**
Typical banana-shape particle size spectrum observed in the static mode of an ozonolysis
experiment is shown in Figure 2a. In the presence of room light, new particles larger than 14 nm
were observed only 58 minutes after the injection of ozone flow. This relative long time is due to the
build-up of $O_3$ concentration and subsequent accumulation of oxidation products. No particles were
formed in the absence of room light or ozone. In the dynamic mode experiments, $O_3$ in the bag
reactor was kept at its maximum concentration 200 ppbv. With a prolonged residential time of 67
min, the particles grew to 102±23 nm, which was measured by the SMPS at the outlet of the bag
reactor. The TOC and TI measurements show that organic compounds contributed more particle
mass than iodine with TOC/(I+TOC) ratio of 96.1±2.9% (Table 1).
In the OH-enhanced experiment (dynamic mode), more particulate products were generated with
enhanced oxidation capacity: TI in the particles increased by a factor of 10.8; TOC increased by a
factor of 2.7; particle number concentration increased by a factor of 7.4. On the other hand, particle
size decreased to 73 nm and TOC/(TI+TOC) ratio decreased to 92.9% (Table 1). These differences
indicate that more iodine nuclei were produced with enhanced OH concentration, probably via
OIO+OH$\rightarrow$ HOIO$_2$ (Plane et al., 2006). Competitive uptake of condensing organic vapors onto these
iodine nuclei limited the growth of individual new particles. Nevertheless, organic compounds
overwhelmingly dominated over iodine in the mass contribution to new particle growth.
The significant organic contribution observed in the laboratory condition is generally consistent
with TOC/(I+TOC) ratio of 98.2% in 10-56 nm new particles collected during a coastal I-NPF event in
China (Yu et al., 2019), although TOC and TI during the field event are two orders of magnitude
lower than those in the bag reactor (Table 1). Mean diameter of new particles was observed to be
only 16 nm during the field event. But those small new particles are expected to grow into CCN
active sizes, given longer residence time and uptake of more condensing vapors in the atmosphere.
**3.2 Macroalgal emission**
It is of particular interest to know what VOCs are emitted from coastal macroalgae. They are



potential precursors of iodine particle nucleation and growth. The canister sampling followed by
GC-MS analysis showed that the top 9 non-CHO compounds with highest TIC peak areas (Table 2)
are $C_5$ alkanes, $C_{10}$ alpha-pinene and halogenated $C_1$, $C_3$ and $C_5$ alkanes. The top 10 CHO
compounds are $C_2$-$C_6$ alcohols and carbonyls with saturated or unsaturated carbon chain.
Iodide-CIMS is more sensitive to more oxygenated or acidic compounds and thus complementary
to the GC-MS measurement. The 76 organic precursors detected by iodide-CIMS before ozone
addition were characterized by $C_{1,2,3,6}$ and $O_{2-3}$ formulas (Figure 3a). The top 7 compounds with
highest ion intensities were $CH_2O_2$, $C_2H_4O_2$, $C_3H_6O_3$, $C_6H_{10}O_3$, $C_2H_6O_2$, $C_4H_8O_2$ and $C_6H_{12}O_3$,
which accounted for 82.5% of total ion intensity. They are $C_1$-$C_6$ mono-carboxylic acids, hydroxyl
carboxylic acids or oxo-carboxylic acids with 2 to 3 oxygen atoms (Table 2). Their carbon atom
numbers are in general consistent with the VOCs detected by GC-MS.
Relatively high signals of $HNO_3$ were observed as $NO_3^-$ and $HNO_3I^-$ before the addition of ozone
to the bag reactor. Because $HNO_3$ and $HNO_2$ were also observed as deprotonated ions or $I^-$ clusters in
the particle phase during the NPF, $HNO_3$ is also an important precursor of particle formation.
**3.3 Gaseous and particulate products**
**3.3.1 Inorganic molecules and radicals**
Being different from nitrate-CIMS, our iodide-CIMS did not detect nucleating clusters of iodine
oxides or oxyacids after the addition of ozone. Instead, dozens of new inorganic molecules or
radicals were observed as clusters with $I^-$, $NO_3^-$ or deprotonated ions in the gas or particle phase
(Figure 4). We grouped these species by elemental composition and investigated their role in the
NPF by observing how their gaseous ion intensities evolved during the NPF event in the bag reactor
(Figure 2b-2f).
1. Cl, I, $Cl_2$ and ClI in the gas phase: the intensities of I and Cl increased ca. 10 minutes before 14
nm particles appeared and decreased as the particles grew up. Based on prior work of Burkholder et
al. (2004); Jimenez et al. (2003); O'dowd et al. (2004), we suggested the photolysis of $CH_2Cl_2$,
$CHBrCl$, $CH_3I$ and $C_3H_7I$ was the source of halogen atoms (e.g., $CH_3I+h\nu \rightarrow CH_3+I$). There was a
time lag of 20-25 minutes between the appearances of Cl and I and those of ClI and $Cl_2$, which were
probably from the recombination of Cl and I atoms.
2. $IO_2$, IO, ClIO, $INO_2$ and $ClNO_2$ in the gas phase: these species showed a similar time series to I
and Cl atoms. IO, $IO_2$ and ClIO could be from the reactions between I, ClI and $O_3$. $INO_2$ is usually
thought to form upon the reaction $I+NO_2+M \rightarrow INO_2+M$ (Saiz-Lopez et al., 2012). $ClNO_2$ was likely





to form upon similar reaction between Cl and $NO_2$ in the bag reactor.
3. $HIO_3$ and $INO_3$: the two species seem to be the end products of above intermediates, because
their intensities kept on increasing during new particle growth. $INO_3$, which is iodine nitrate $IONO_2$,
was detected in both gas and particle phases. $IONO_2$ probably formed upon the recombination of IO
and $NO_2$ ($IO+NO_2+M \rightarrow IONO_2+M$) (Saiz-Lopez et al., 2012). $HIO_3$ was likely to form via
$OIO+OH \rightarrow HOIO_2$ or $I + H_2O + O_3 \rightarrow HOIO_2 + OH$ (Plane et al., 2006; Martín et al., 2020). $HIO_3$
was not detected in particle phase by iodide-CIMS, which is contrary to the observation by
HPLC-ICP-MS that total iodine was mostly dominated $IO_3^-$ peak. The signals of $IO^-$, $IO_2^-$ and
$HIONO_3^-$ in the particle phase are therefore most likely to result from thermal decomposition of
$HIO_3$ to HIO and $HIO_2$ in the FIGAERO thermal desorption process.
4. $CH_3SO_3H$, $S_2^-$, $S_3^-$, $SO_3^-$: We observed methane sulfonic acid ($CH_3SO_3H$, MSA) in both gas and
particle phases. Gaseous MSA increased in the beginning, but decreased after new particles appeared
(Figure 2f). Apparently, our measurement suggested MSA contributed to the growth of new particles,
but it is unknown if it also participated in nucleation. We suggested $S_2^-$, $S_3^-$, $SO_3^-$ ions observed in the
particle phase were thermal decomposition products of MSA.
**3.3.2 Gaseous organic products**
After ozone addition, a gradual transformation from $C_1$-$C_3$ precursors to $C_5$-$C_8$ gaseous products
was observed during the NPF process (Figure 2h). In the meanwhile, the oxygen atom number of the
compounds increased from 2-3 to 4-7 (Figure 2g). The formation of compounds with more carbon
atoms than the parent VOCs is unlikely in the gas phase, except bimolecular reactions of stabilized
Criegee intermediates (SCIs) that typically form upon alkene ozonolysis. Similar to isoprene
ozonolysis   (Riva et al., 2017; Inomata et al., 2014), we propose the SCI addition mechanism can
also explain the transformation observed in our case: (1) $C_4$ SCIs formed upon the ozonolysis of
CHO precursors with C=C double bonds (e.g., those observed by GC-MS in Table 2). (2) the
insertion of $C_4$ SCIs into carboxylic acid precursors (e.g., those observed by CIMS in Table 2)
produced oligomeric hydroperoxides. An example was shown in Scheme I for the reactions of most
abundant ethyl vinyl carbinol ($C_5H_{10}O$), ozone and formic acid ($CH_2O_2$), but the same mechanism is
also applicable for ethyl vinyl ketone ($C_5H_8O$) and other abundant $C_2$-$C_5$ carboxylic acids and
hydroxyl carboxylic acids. As a result, a series of gaseous oligomeric hydroperoxides $C_5H_{10}O_5$,
$C_6H_{10}O_5$, $C_6H_{12}O_5$, $C_7H_{12}O_6$, $C_7H_{14}O_6$, $C_8H_{14}O_5$, $C_8H_{16}O_6$, $C_8H_{16}O_5$ and $C_9H_{16}O_6$ were observed with
high intensity by iodide-CIMS.



220            Scheme I

### 3.3.3 Particulate organic products

In the end of a typical ozonolysis experiment (dynamic mode), 100 and 364 new formulas were
observed in the gas and particle phases, respectively, including 73 semi-VOCs appeared in both gas
and particle phases. Those semi-VOCs accounted for 81 and 20% of total ion intensities of gaseous
and particulate products, respectively. Being different from unimodal atom number distributions of
gaseous products ($C_{max}$= 7 and $O_{max}$=5, Figure 3b), particulate products were characterized by
distinct bimodal or trimodal distribution of carbon number ($C_{max}$=8, 14 and 16, Figure 3c) and
oxygen number ($O_{max}$=4 and 8), implying possible dimer formation via accretion reactions in the
particle phase.
ESI-Orbitrap MS differs from FIGAERO-iodide-CIMS in extraction method (ultrasonic solvent
extraction from quartz fiber filter *vs.* thermal desorption from PTFE membrane filter), ionization
source (electrospray ionization *vs.* iodide-adduct chemical ionization) and MS resolving power
(70000 *vs.* 4500). The result showed that ESI-orbitrap MS and FIGAERO-iodide-CIMS detected 336
and 364 organic formulas, respectively, in the particle phase. 167 organic formulas were commonly
observed by both methods, which accounted for 87% and 54% of total ion intensity of organic
formulas by the two methods, respectively (Figure S1). As shown in Figure 3c and 3d,
FIGAERO-iodide-CIMS had better sensitivity toward the organic compounds with more oxygen
atoms (e.g., O⩾8) and carbon atoms (e.g., C⩾10). As a result, bimodal carbon and oxygen atom
number distributions were observed by FIGAERO-iodide-CIMS, but not ESI-orbitrap MS.
The measurement by ESI-orbitrap MS provided more insights about the formation mechanism of
particulate products. We compared the 336 formulas detected by ESI-orbitrap MS in our study with
the 414 formulas of isoprene ozonolysis SOA products (Nguyen et al., 2010) and 922 formulas of
alpha-pinene ozonolysis SOA products (Putman et al., 2012) measured by the ESI-orbitrap MS. It
was found that 72% of the formulas in this study can also be found in isoprene SOA, but only 39%
can be found in alpha-pinene SOA. This seems to imply that some similar alkene ozonolysis
reactions occurred in our system and isoprene ozonolysis.
For such a highly complex system full of various algae-emitted precursors, it is impossible to



simply propose a reaction mechanism to explain the formation of all particulate products, nor to list
all reactions occurring in the bag reactor. On the basis of particle-phase oligomer chemistry (Seinfeld
and Pandis, 2016), especially the well-understood isoprene ozonolysis SOA chemistry (Nguyen et al.,
2010; Inomata et al., 2014; Riva et al., 2017), we suggest a variety of accretion reactions without
uniform oligomerization pattern (e.g., esterification, aldol condensation, hemiacetal reactions,
peroxyhemiacetal formation and SCI reactions, etc.) transformed $O_{max}=4$ and $C_{max}=8$ multifunctional
monomers (like alcohols, carbonyls, hydroperoxides, carboxylic acids) to $O_{max}=8$ and $C_{max}=14$ or 16
dimers. Scheme II illustrated addition type self- and cross-oligomerization between $C_6$ and $C_8$
monomers produces $C_{14}$ and $C_{16}$ dimers. All the formulas in Scheme II are among the most abundant
ones observed in the particle phase by the iodide-CIMS.
$$C_6H_{6-12}O_{3-6}+C_8H_{10-16}O_{3-6} \rightarrow C_{14}H_{16-26}O_{6-12}$$
$$C_8H_{10-16}O_{3-6}+C_8H_{10-16}O_{3-6} \rightarrow C_{16}H_{20-32}O_{6-12}$$ Scheme II
**4. Conclusions**
Using a suite of mass spectrometers, we reported, for the first time, the chemical compositions of
volatile precursors emitted by real-world coastal macroalgae and their gaseous and particulate
oxidation products. In the presence of room light and ozone, the photolysis of halogenated $C_{1,3,5}$
alkanes ends up as $HIO_3$ and $INO_3$. It was most likely $HIO_3$ initiated NPF and provided nuclei for the
further condensation of other products like $INO_3$, MSA and CHO compounds. Gas-phase SCI
reactions and particle-phase accretion reactions transformed $C_1$-$C_6$ and $O_2$-$O_3$ precursors gradually to
particulate products with $C_{max}=8$, 14 and 16 and $O_{max}=4$ and 8. As a result, organic carbon were
found to overwhelmingly dominated over iodine in the mass contribution to the new particle growth.
Although our instruments did not allow the detection of nucleating clusters of iodine oxides or
oxyacids, our study provided important complementary information to the ongoing laboratory and
field researches of coastal I-NPF.
**Data Availability**
All data related to figures and tables in this study are archived and made available through
Zenodo data repository https://doi.org/10.5281/zenodo.6965859.
**Financial support.**
This work was supported by the National Science Foundation of China (grant no. 41975831 and
42175131) and Start-up research funding from China University of Geosciences.



**Competing Interests.**
The authors declare that they have no conflict of interest.
**Author contributions**
HY designed the experiment. YW, XH and CX conducted the experiments. YW and HY analyzed
the data and wrote the manuscript. QW and XG reviewed and revised the manuscript.

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





Table 1. Particle number concentration ($N$), mean diameter ($D_p$), total organic carbon (TOC) and total iodine (TI) of new particles with a residential time of 67 min in the bag reactor in the ozonolysis experiments and OH-enhanced experiment (dynamic mode). Those of 10-56 nm new particles collected by a nano Micro-Orifice Uniform Deposit Impactor (nano-MOUDI, MSP, Inc.) during an I-NPF event at a coastal site of Ningbo, China (Yu *et al.*, 2019) were also listed.

| | TOC (µg m⁻³) | TI (µg m⁻³) | TOC/(TI+TOC) | $N$ (cm⁻³) | $D_p$ (nm) |
|---|---|---|---|---|---|
| ozonolysis experiments | 45.6±9.7 | 0.88±0.34 | 96.1±2.9% | $(5.58\pm2.04)\times10^4$ | 102±23 |
| OH-enhanced experiment | 125.3 | 9.5 | 92.9% | $4.16\times10^5$ | 73 |
| I-NPF event at a coastal site of China | 0.7 | 0.0135 | 98.2% | $6.00\times10^5$ | 16 |





Table 2. Major volatile organic compounds emitted by macroalgae as potential NPF precursors, sorted by TIC peak area measured by GC/MS or MS peak intensity measured by iodide-CIMS

| | Formula | Structure | Peak area/MS peak intensity |
|---|---|---|---|
| 1 | $C_5H_{12}$ | | $1.90\times10^6$ |
| 2 | $C_5H_{10}$ | | $1.59\times10^6$ |
| 3 | $CH_3I$ | | $1.37\times10^6$ |
| 4 | $C_3H_7I$ | | $7.60\times10^5$ |
| 5 | $CHBr_3$ | | $4.71\times10^5$ |
| 6 | $C_5H_{11}I$ | | $3.75\times10^5$ |
| 7 | $CHBr_2Cl$ | | $2.71\times10^5$ |
| 8 | $CH_2Cl_2$ | | $2.55\times10^5$ |
| 9 | $C_{10}H_{16}$ | | $2.26\times10^5$ |
| 1 | $C_2H_6O$ | | $1.70\times10^7$ |
| 2 | $C_3H_6O$ | | $1.38\times10^7$ |
| 3 | $C_4H_6O_2$ | | $1.30\times10^7$ |
| 4 | $C_6H_{12}O$ | | $1.03\times10^7$ |
| 5 | $C_5H_{10}O$ | | $1.00\times10^7$ |
| 6 | $C_4H_{10}O$ | | $5.16\times10^7$ |
| 8 | $C_2H_4O$ | | $3.46\times10^7$ |
| 9 | $C_6H_{10}O$ | | $2.88\times10^7$ |
| 7 | $C_5H_8O$ | | $1.45\times10^7$ |
| 10 | $C_4H_8O$ | | $1.37\times10^7$ |
| 1 | $CH_2O_2$ | | $1.58\times10^6$ |
| 2 | $C_2H_4O_2$ | | $9.52\times10^5$ |
| 3 | $C_3H_6O_3$ | | $9.21\times10^5$ |
| 4 | $C_6H_{10}O_3$ | | $4.44\times10^5$ |
| 5 | $C_2H_6O_2$ | | $2.88\times10^5$ |
| 6 | $C_4H_8O_2$ | | $1.17\times10^5$ |



| 7 | $C_6H_{12}O_3$ | | $1.12 \times 10^5$ |



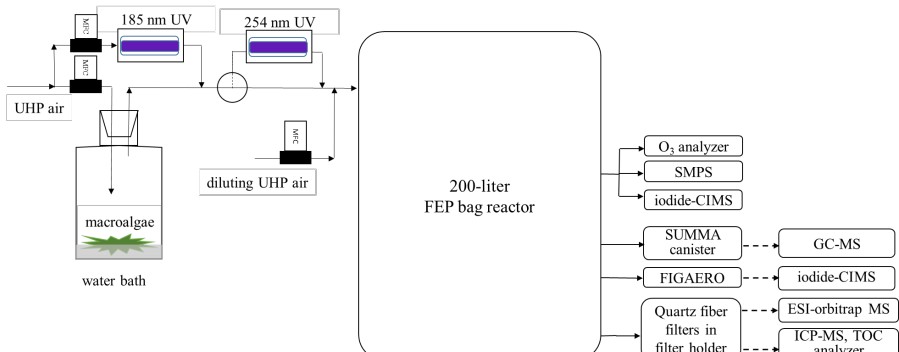

Figure 1. Schematic of experimental setup. Solid line: air flows. Dashed lines: sent for offline chemical analysis.



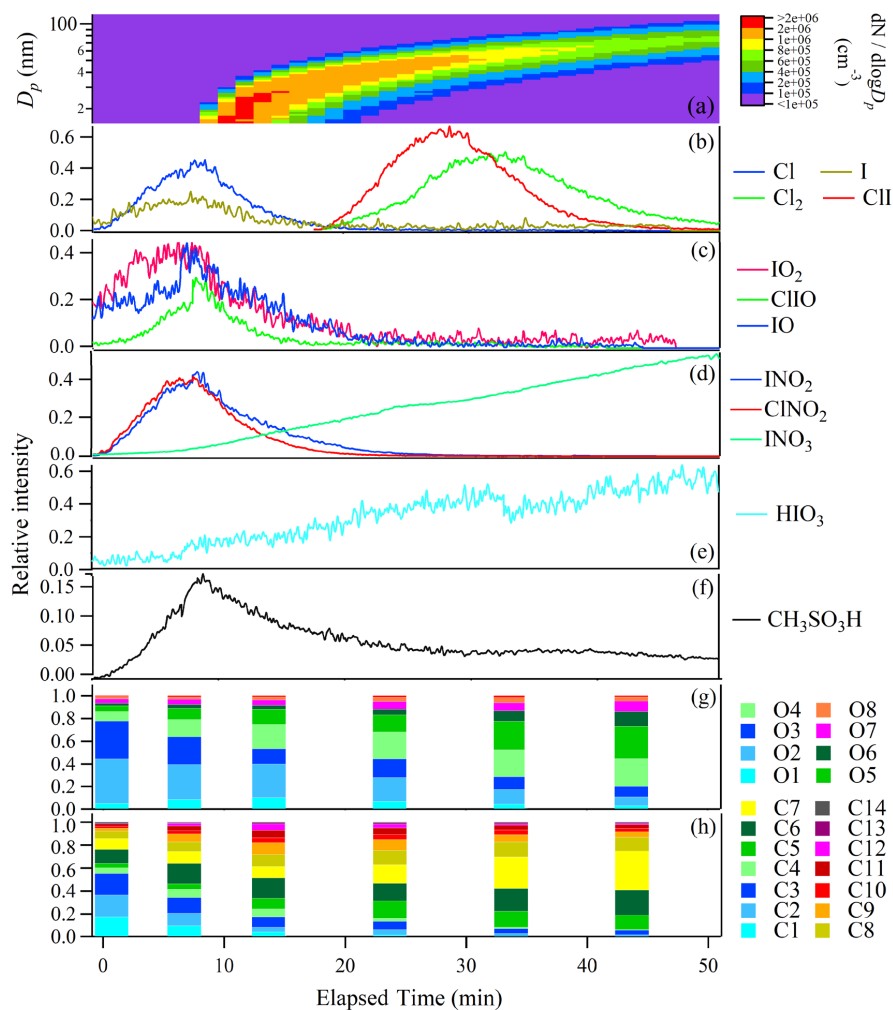

Figure 2. Time evolution of particle number size distribution (a) and relative intensities of gaseous molecules and radicals (b-f); the fractions of organic compounds grouped by O and C atom numbers in the selected time points (g-h) in a typical ozonolysis experiment (static mode). Time zero was chosen as the start time when $HIO_3$ was observed.





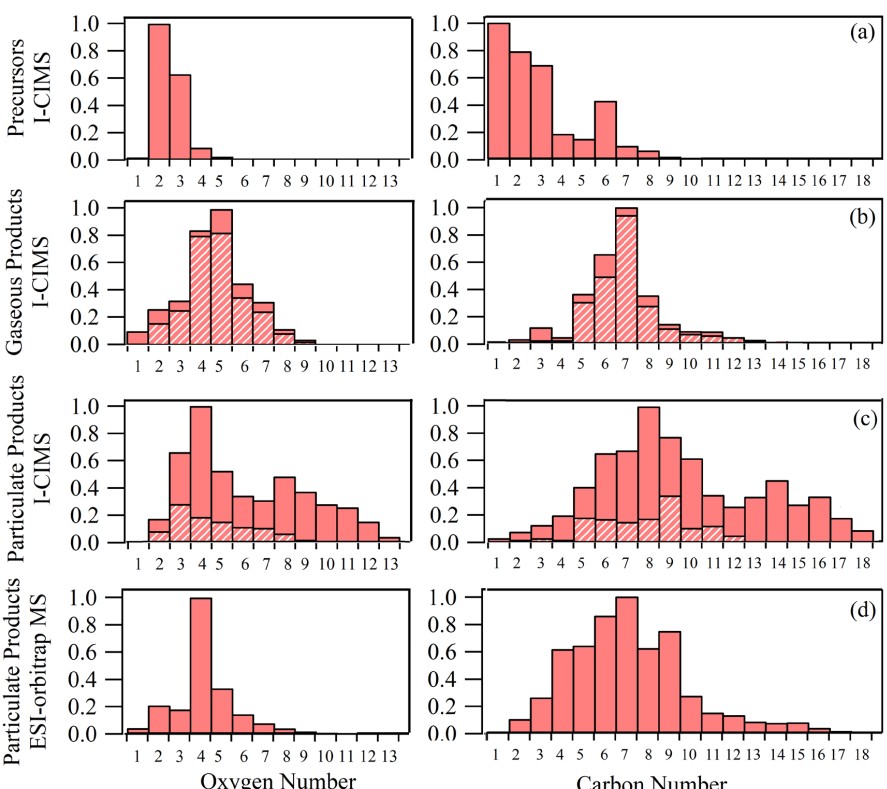

Figure 3. Oxygen and carbon atom number distributions of potential VOC precursors (a), gaseous products (b) and particulate products measured by iodide-CIMS (c), as well as the particulate products measured by ESI-orbitrap MS (d) in a typical ozonolysis experiment (dynamic mode). Hatched bars indicate the fractions of organic formulas observed in both gas and particle phases by iodide-CIMS.





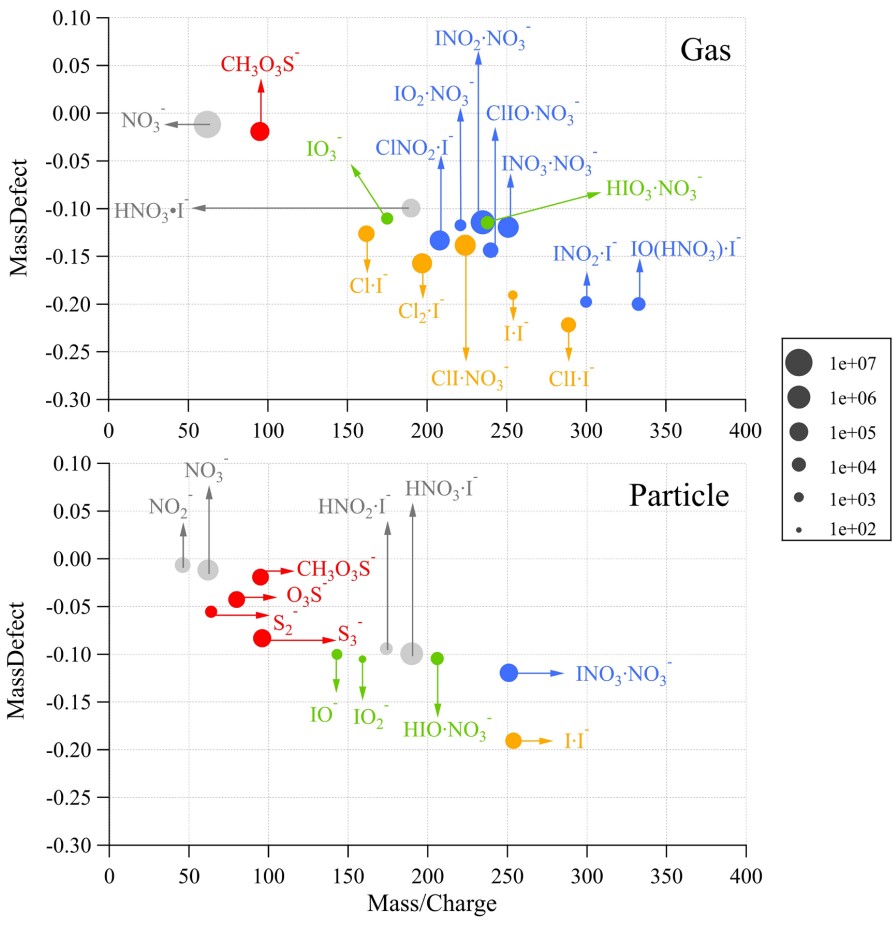

Figure 4. Integrated ion intensities of inorganic molecules and radicals in the gas phase (static mode) and particle phase (dynamic mode) measured by iodide-CIMS in a typical ozonolysis experiment. The ions were coded in color according to their elemental composition