# Peer review of "Chemical characterization of organic compounds involved in iodine-initiated new particle"

_EGUsphere, 2022_

## Author Comment (AC1)

**A point-to-point response and relevant changes made in the revised manuscript**

**community comments**

Wan et al. have performed an interesting laboratory study showing that the organic compounds co-emitted with iodine bearing molecules by exposed tidal macroalgae dominate particle growth in iodine-triggered NPF events. The chemical evolution of the emitted organic precursors is investigated by means of iodide-CIMS, showing that alkene ozonolysis and criegee intermediate gas-phase reactions and particle-phase accretion reactions increase the number of carbon and oxygen atoms of the organic compounds observed. Some information about inorganic iodine molecular cluster precursors is also obtained.

I have listed a few comments below that the authors may want to consider to improve their manuscript.

Page 2, line 35. Here and elsewhere: Mart *n* et al., 2020 -> G ómez Mart *n* et al., 2020

Re: corrected.

Page 2, line 38. A previous study also examined the growth of iodine oxide clusters in the presence of condensable vapours such as H2SO4 or oxalic acid (Saunders et al., 2010)

Page 2, line 47. An opening sentence indicating that organic compounds have been observed in particles formed in I-NPF events (Vaattovaara et al., 2006; Yu et al., 2019) would be useful.

Re: we add the citations in line 46-50:

"Organic compounds have also been suggested to be involved in coastal NPF (Vaattovaara et al., 2006; Yu et al., 2019). Huang et al. (2022) and Saunders et al. (2010) investigated the effect of uptake of meso-erythritol, glyoxal, dimethylamine and oxalic acid on the growth of iodine oxide nanoparticles. However, no prior work has investigated the exact chemical identity of organic compounds (other than iodomethane) and their role in I-NPF."

Page 2, line 48. More common names for this compound are iodomethane and methyl iodide

Re: iodomethane is used now.

Page 3, line 81. Indicate in this paragraph an estimate of water vapour concentration or RH in the experiment.

Re: we add in line 86-87:

"*RH* was estimated to be 10% in the bag reactor assuming 0.3 lpm water-saturated VOC flow was diluted by 2.7 lpm dry air flow."

Page 3, line 84. Note that even though up to 10% of O1D may end up as OH under atmospheric conditions, the rest will be quenched to O3P, and that O3P reacts both with I2 and iodomethane to make IO. Moreover, it is well known that OH reacts quickly with I2 to make HOI. Therefore, in these experiments additional photolytic sources of IO are present, plus a source of HOI. This may obscure the interpretation of the "OH-enhanced" experiments.

Re: we add the following description (line 89-93, 245-247) to explain the oxidants in the OH enhanced experiment.

"OH radicals were produced via the reaction  $O_3 + hv \rightarrow O_2 + O({}^1D)$  and  $O({}^1D) + H_2O \rightarrow 2OH...$ Other oxidants may include  $O({}^3P)$  resulted from the quenching of  $O({}^1D)$  (Li et al., 2015)."

"These differences indicate that more iodine nuclei were produced with enhanced oxidation capacity, probably via  $OIO+OH \rightarrow HOIO_2$  (Plane et al., 2006) and  $O(^{3}P)+CH_{3}I \rightarrow IO+CH_{3}$  (Teruel et al., 2004)."

In the "OH-enhanced" experiments, we showed bulk TOC and TI results only, interpreting that more iodine nuclei and particulate products were generated with enhanced oxidation capacity, while organic compounds still overwhelmingly dominated over iodine in the mass contribution to new particle growth. These interpretations were not obscured by more complicated oxidant species.

Page 4, line 117. Indicate ultrasonication time and power.

Re: We add in line 127:

"Ultrasonication time and power were 20 minutes and 150 Watt."

Page 5, line 145. It is likely that this effect is rather due to the presence of ground state oxygen atoms in the flow. O3P will free additional iodine atoms by reaction with I2 and CH3I.

Re: We add in line 245-247:

"These differences indicate that more iodine nuclei were produced with enhanced oxidation capacity, probably via  $OIO+OH \rightarrow HOIO_2$  (Plane et al., 2006) and  $O({}^3P)+CH_3I \rightarrow IO+CH_3$  (Teruel et al., 2004)."

Page 6, line 171. Is HNO3 then emitted by algae? I think explaining a bit more about the source of HNO3 is necessary, since it indirectly allows detection of most of the inorganic compounds reported. In fact the iodide CIMS in practice operates in these experiments as a nitrate CIMS for inorganic iodine compounds.

Re:  $NO_3^-$  sometimes exists as contaminant in the CIMS system, probably from air leak or contaminated air supply tubes of the CIMS. The system has to be cleaned before each experiment to eliminate background contaminant. Therefore, elevated  $NO_3^-$  signal did not surprise us very much when the sample air was from the bottle containing "dirty" algae and natural seawater.

In line 161-164, we add

"Relatively high signals of  $NO_3^-$  and  $HNO_3\Gamma$  were observed before the addition of ozone to the bag reactor. They were likely  $HNO_3$  or nitrate vaporized from algal specimens or natural seawater. Because  $NO_3^-$  and  $HNO_3\Gamma$  were also observed in the particle phase during the NPF (Figure 4), we assume  $HNO_3$  was also an important precursor of particle formation."

Page 6, line 174. While iodide CIMS maybe a good technique for detecting organics, it is probably not be the best technique for detecting inorganic iodine compounds, for the obvious reason that the source

of charge is the iodide anion, which may obscure the interpretation of the observed ion clusters. No discussion of this potential interference has been included in this paper.

Re:

We add the following discuss in line 209-217:

"We noticed that  $CH_3I$  vapor was added as ion source reagent to the ion molecule reactor (IMR) of iodide-CIMS. It is likely that this extra  $CH_3I$  in the IMR might obscure the interpretation of the observed iodine containing clusters. We believed that ion source reagent  $CH_3I$  should have relatively small interference with inorganic iodine compounds from the bag reactor, on the basis of 2 facts: (1) ion source reagent  $CH_3I$  was added directly from permeation tube into the IMR. Without photolysis, ion source reagent  $CH_3I$  in the IMR should not become a source of I and  $I_xO_y$ . (2) the concentration of ion source reagent  $CH_3I$  and its potential products should be quite constant as long as  $O_3$  was present in the IMR, which was not supported by the variable signals of I, CII, IO2, IO, CIIO, HIO3, INO2 and INO3 in Figure 2."

Page 6, line 177. Alongside Figure 4, it would be very useful showing a table with the correspondences between observed anions and proposed parent neutral molecules. Such correspondence is not always straightforward, as we have argued recently (Gómez Mart ń et al., 2022).

Re:

Thank you for pointing this out. Ion clusters were labeled alongside the signal dots of Figure 4. For each ion cluster, parent neutral molecule is on the left hand side of middle dot, while the clustering ion  $I^{\circ}$  or NO3- is on the right hand side. Those without a clustering ion are shown as bare anions.

These descriptions are now added to Figure 4 caption.

Page 6, line 182. What about I2 and HOI photolysis? Why are you ruling out I2 and HOI as iodine sources?

Re: we did not include  $I_2$  and HOI because they were not observed in the gas flow by either GC-MS or I-CIMS.

In line 175-178, we add

"we suggested the photolysis of  $CH_2Cl_2$ , CHBrCl,  $CH_3I$  and  $C_3H_7I$  was the source of halogen atoms (e.g.,  $CH_3I+hv \rightarrow CH_3+I$ ), although we could not exclude the photolysis of other precursors like  $I_2$  and HOI that are invisible to GC-MS and I-CIMS."

Page 6, line 184. Unlikely. Much faster reactions are:

Cl+I2->ICl+I

Cl+ICl->Cl2+I

The time traces in Figure 2b are qualitatively consistent with this sequence of reactions

Re: thank you for pointing it out.

In line 178-180 we update to:

"There was a time lag of 20-25 minutes between the appearances of Cl and I and those of ClI and Cl2, which were probably resulted from anion exchange reactions of Cl I and II with Cl atoms."

Page 6, line 188. These experiments employ UHP air. What is then the source of NO2 in this system? There is no easy route from HNO3 to NO2.

Page 7, line 193. Again, what is the source of NO2 in this system? This must be discussed, since you are concluding that IONO2 is contributing to particle growth. In our recent work on the nitrate CIMS system in the context of I-NPF (Gomez Martin et al., 2022), we have found that IO3-, HIO3.NO3- (or rather HNO3.IO3-) and IONO2.NO3- are products of the reaction between NO3- and I2O3. I am skeptical about the presence of IONO and IONO2 in this system because of the unlikely presence of NO and NO2, and I suspect that IONO.NO3- and IONO2.NO3- could be products of IxOy+NO3- also in these experiments.

Re: thank you for the comments on IONO and IONO2. Now we cite Gomez Martin et al., 2020 and 2022 and add the following paragraph in line 185-193.

3.  $INO_2$ ,  $CINO_2$  and  $INO_3$ :  $INO_2$  and  $CINO_2$  were detected in gas phase with similar time evolution with halogen atoms and halogen oxides.  $INO_3$  was found in both gas and particle phases.  $INO_2$  and  $INO_3$  were usually thought to form upon the reactions  $I+NO_2+M \rightarrow IONO+M$ and  $IO+NO_2+M \rightarrow IONO_2+M$  in the atmosphere (Saiz-Lopez et al., 2012), which seems to be unlikely in our bag reactor because  $NO_2$  was not added. Considering  $NO_3^-$  was ubiquitous in the bag reactor of our experiment, it is likely that  $INO_2$  and  $INO_3$  formed via  $I_2O_2+NO_3^- \rightarrow$  $IO_3^-+IONO$  and  $I_2O_3+NO_3^- \rightarrow IO_3^-+IONO_2$ . These reaction pathways have been supported by theoretical calculation and flow tube mass spectrometry experiments (G ómez Mart ń et al., 2022; G ómez Mart ń et al., 2020).  $CINO_2$  was likely to form upon similar reaction between  $Cl_2O_2$  and  $NO_3^-$  in the bag reactor.

Page 7, line 193. Following my previous comment, at least part of the signal attributed to HIO3 results from I2Oy+NO3- (Gómez Mart ń et al., 2022)

Page 7, line 194. Note that Gomez Martin et al., 2020 never argued in that HOIO2 would form from I + H2O + O3 -they rather argued the opposite. The source of HOIO2 remains to be confirmed, although the reaction between I2O5 and the water dimer is currently our best candidate, where I2O5 would be a photolysis product of a higher iodine oxide (Gómez Mart n et al., 2022).

Page 7, line 196. This is in disagreement with the observations by He et al. 2022 using a Br-CIMS FIGAERO. They did observe HIO3 in the particles. This disagreement should be discussed.

*Heating of HIO3 between 100* °C *and 200* °C *results in dehydration and formation of I2O5 (Selte and Kjekshus 1968), so the desorption temperature in is critical.*

The IO- and IO2- signals may be secondary products of the reaction between I2O5 and I-.

Re: thank you for the comments on HIO, HIO2 and HIO3. The following paragraph is updated in line 194-203:

3. HIO,  $HIO_2$  and  $HIO_3$ :  $HIO_3$  seems to be the end product of above intermediates, because its gas-phase ion intensity kept on increasing during new particle growth. Based on this fact, we

assume that HIO3 could be from  $I_2O_5+H_2O \rightarrow 2HIO_3$  or  $I_2O_y+NO_3^- \rightarrow IO_3^-+INO_y$ . On the other hand, HIO3 was not detected in particle phase by iodide-CIMS, which is contrary to the offline analysis of quartz filter by HPLC-ICP-MS showing that total iodine was mostly dominated by  $IO_3^-$  peak. We speculate that HIO3 might have been dehydrated to  $I_2O_5$  under thermal desorption temperature up to 180 °C in FIGAERO. The signals of  $IO^-$ ,  $IO_2^-$  and HIONO3- (corresponding to HIO and HIO2) were found in particle phase, but not in gas phase. He et al. (2021) proposed HIO2 formation via  $\Gamma + H_2O + O_3 \rightarrow HIO_2$  or  $I_2O_2 + H_2O \rightarrow HIO + HIO_2$ . With limit experimental evidence of our work, the exact formation pathways of HIOx remains to be explored in future.

---

## Author Comment (AC2)

**A point-to-point response and relevant changes made in the revised manuscript**

**Anonymous Referee #1**

*Manuscript entitled "Chemical characterization of organic compounds involved in iodine-initiated new particle formation from coastal macro-algal emission" studied the identity and transformation mechanisms of organic compounds from low-tide macroalgal emission. This manuscript simulates vapor emission oxidation and new particle formation (NPF) experiments of real coastal macroalgae in a bag-reactor. Based on the integrated mass spectrometry measurements, the authors report for the first time a variety of volatile precursors and their oxidation products in the gas and particle phases in such a highly complex system. The results show that organic compounds dominate the growth of new particles induced by iodine species.*

*This paper falls in an active field of research, and I believe it brings interesting insights to the study of the ongoing laboratory and field researches of coastal I-NPF. The paper could be published in the journal assuming some minor corrections.*

> **Re:** We thank the referee for careful examination and important comments of the manuscript. Point-to-point responses were given below. Changes were made in the revised manuscript accordingly.

*Comments*

*Page 4 Lines 96-98: What is the proportion of VOCs and O3 in the VOCs/O3 flow of the dynamic mode, respectively?*

> **Re:** we used the vapor mixture emitted from macroalgae in the experiment. As one can see, there are many VOC species in the mixture. Unlike those studies focusing on a single precursor, it is not possible for us to estimate the ratio of VOCs and $O_3$. The purpose of this study is to qualitatively measure the oxidation products of algae-emitted VOCs in the simulated NPF event.

*Page 4 Lines 98-99: Does the residence time of 67min refer to the sampling time of particulate matter?*

> **Re:** No, residence time is reaction time in the bag reactor, which is the volume of bag reactor divided by the flow rate of VOCs/$O_3$ flow. Now we add in line 103-104:

> > *"The bag reactor was then operated in a dynamic mode for a few hours to collect enough particles for offline chemical analysis."*

*Page 5 Lines 133-134: How can we see from Figure 2a when O3 is injected? And when to add light? Why only see the figure of SMPS under static conditions of the ozonolysis experiment.*

> **Re:** our pilot study showed that no particles formed in the absence of room light or $O_3$. Therefore, $O_3$ and light were supplied throughout the experiments of chemical characterization. Even in the first 48 minutes after $O_3$ were added, we could not observe any particle or gas-phase product. So we

did not show the data during that early period. Time zero of Figure 2 was set as the time when gaseous products first appeared.

In the dynamic condition of the experiments, continuous time evolution of particle size can not be measured. So we did not show SMPS particle size spectrum. The particle size at the outlet of the bag reactor was shown in Table 2.

Now we add the following description in line 142-147:

*"No particles formed in the absence of room light or $O_3$. Therefore, light was on throughout the experiments reported in the article. In the static mode experiments, we could not observe gas-phase products until 48 minutes after $O_3$ injection. New particles larger than 14 nm were observed only 58 minutes after $O_3$ injection. Afterwards, new particles begun to grow to form a typical banana-shape particle size spectrum (Figure 2a). This relatively long waiting time was likely due to the build-up of $O_3$ and oxidation products. Time zero of Figure 2 was thus set as the time when gaseous products first appeared."*

*Page 5 Line 157: Please indicate what kind of macroalgae you choose and how to preserve the algae and seawater. And why you choose this type of macroalgae?*

**Re**: we have the description in line 77-78:

*"Undaria pinnatifida, a common brown seaweed species at Xiangshan gulf of east China coast, was collected from local intertidal zone and stored at $-10^{o}C$ until the experiments."*

*Page 8 Lines 225-229: Does accretion reactions or dimer formation change particle size? Please describe the accretion reaction in detail.*

**Re:** As one can see in Figure 2a, new particles kept on growing to form a typical banana-shape particle size spectrum. However, we can not differentiate, by our experiment, the growth was due to the uptake of gaseous products or accretion reactions in the particles. In theory, accretion reaction, by itself, should not brought new mass to the particles.

We described the accretion reactions in line 284-298:

*"For such a highly complex system full of various algae-emitted precursors, it is impossible to simply propose a reaction mechanism to explain the formation of all particulate products, nor to list all reactions occurring in the bag reactor. On the basis of particle-phase oligomer chemistry (Seinfeld and Pandis, 2016), especially the well-understood isoprene ozonolysis SOA chemistry (Nguyen et al., 2010; Inomata et al., 2014; Riva et al., 2017), we suggest a variety of accretion reactions without uniform oligomerization pattern (e.g., esterification, aldol condensation, hemiacetal reactions, peroxyhemiacetal formation and SCI reactions, etc.) transformed $O_{max}=4$ and $C_{max}=8$ multifunctional monomers (like alcohols, carbonyls, hydroperoxides, carboxylic acids) to $O_{max}=8$ and $C_{max}=14$ or 16 dimers. As an example, we used the following two reaction equations to illustrate addition type cross-oligomerization between $C_6$ and $C_8$ monomers and self-oligomerization of $C_8$ monomers, respectively:*

$$C_6H_{6-12}O_{3-6}+C_8H_{10-16}O_{3-6} \rightarrow C_{14}H_{16-26}O_{6-12}$$

$$C_8H_{10-16}O_{3-6}+C_8H_{10-16}O_{3-6} \rightarrow C_{16}H_{20-32}O_{6-12},$$

*in which the $C_6$ , $C_8$, C14 and C16 formulas in the equations are among the most abundant ones observed in the particle phase by the iodide-CIMS."*
*"*

*Page 8 Lines 230-232: Why do ESI-Orbitrap MS and FIGAERO-iodide-CIMS use quartz fiber filter and PTFE membrane filter, respectively? Is filter inconsistency the reason why ESI-Orbitrap MS did not measure bimodal distribution?*

**Re:** PTFE membrane filter is the recommended for particle collection and thermal desorption in FIGAERO. Its relative large pore size avoids excessive pressure drop across the filter during the sampling. Quartz fiber filter was used for ESI-orbitrap MS, total iodine and total organic carbon measurements. The advantage of quartz filter is that background contaminants on the filter could be removed using thermal combustion prior to particle collection.

Chemical composition difference observed by ESI-Orbitrap MS and FIGAERO-iodide-CIMS might resulted from many factors, like sample substrate, extraction method, ionization mechanism and MS resolution. Systematic investigation of the effect of these factors is out of scope of this manuscript. The motivation of performing ESI-Orbitrap MS analysis in our work is to facilitate the comparison with isoprene and alpha-pinene ozonolysis products reported by prior ESI-Orbitrap MS measurements in the literature. From our experience, we believe ionization mechanism (ESI *vs.* I⁻ clustering) might be the most important factor.

---

## Author Comment (AC3)

**A point-to-point response and relevant changes made in the revised manuscript**

**Anonymous Referee #2**

The authors reported the chemical composition and evolution of volatile precursors emitted from macro-algae and their oxidation products in the gas and particle phase using a suite of mass spectrometers. But it was shallow and simple about the discussion of the transformation mechanisms of organic compound. I recommend that the authors could make more detailed explanations about the results and explore more precise reaction formulas.

Here are some questions about the methods and results in the following.

**Re**: we thank the referee for careful examination and important comments of the manuscript. Point-to-point responses were given below. Changes were made in the revised manuscript accordingly.

As a response to his/her general comment, we think for such a highly complex system with tens and even hundreds of precursors, it is very difficult for us to present a uniform reaction pattern to explain the complicated interactions among numerous precursors, intermediates and products, or to list all reaction equations that occurred in the bag reactor. Exploring precise reaction equations may be more practical for a one-precursor system. This could be the goal of a future study. More explanation is presented in our response to the last comment.

More descriptions about the formation of inorganic molecules/radicals  $I_xO_y$ ,  $ClNO_x$  and  $HIO_x$  were added in the revised manuscript.

**Method**

**81: "In the three ozonolysis experiments"**

It seems that only one result (without error bar) is shown in this paper. What is about the remaining two experiments?

**Re**: Mean values and standard deviations of total organic carbon, total iodine, particle number and size were presented in Table 1 for the three ozonolysis experiments.

The purpose of this study is to qualitatively identify gas and particle products of algae-emitted VOCs in the simulated NPF event. Because aerosol production TOC and TI were quite constant, we measured chemical composition for only one set of experiment.

**84: "In an additional OH-enhanced experiment"**

The authors conducted this experiment for simulating atmospheric oxidation process, however, you didn't even give the concentration of additional OH and the limitation of the experimental design compared with the real environment wasn't discussed.

**Re:** OH was produced via  $O_3$  photolysis in the UV lamp and consumed via reacting with vapor mixture and wall loss in the bag reactor. So OH concentration changed over time when air flowed through the lamp and the bag reactor. When revising the manuscript, we did SO2 decay experiment in the bag reactor to estimate integrated OH exposure.

The following statement is added in line 90-96

"Integrated OH exposure time was determined by  $SO_2$  decay experiment to be 2.4 days in the experimental apparatus assuming ambient average OH concentration  $1.5 \times 10^6$  molecules cm-3 (see Supporting Material S1). Other oxidants may include  $O({}^3P)$  resulted from the quenching of  $O({}^1D)$  (Li et al., 2015). Because the purpose of this study is to identify gas and particle products of algae-emitted VOCs in the simulated NPF event, significantly higher oxidation level in the bag reactor than atmosphere should not change the conclusions in the article. Wall loss, aerosol yield, reaction rate and other kinetic factors in the bag reactor were also not evaluated."

**In SM Text S2**

**"Integrated OH exposure measurement**

 $SO_2$  decay experiment was conducted to estimate integrated OH exposure in the experimental apparatus, following Lambe et al. 2015. We replaced the macroalgal emission flow by a humidified air flow containing 200 ppbv  $SO_2$  from standard gas cylinder.  $SO_2$  was consumed by the reaction with OH while flowing through 254 nm UV light and the bag reactor.  $SO_2$  mixing ratio at the outlet of the bag reactor was measured with and without UV lamp on, using a Model 43i-TLE  $SO_2$  analyzer (Thermo Scientific Inc.). OH exposure was calculated from the equation

$$OH_{exp} = ln(SO_{2 \ lamp \ off}/SO_{2 \ lamp \ on})/k_{OH+SO2}$$

where  $k_{OH+SO2}$  is 9.49×10-13 cm3 molecule-1s-1 (Burkholder et al. 2015).

 $SO_{2 \text{ lamp off}}$  and  $SO_{2 \text{ lamp on}}$  were measured to be 18 and 13 ppbv, respectively.  $OH_{exp}$  was then calculated to be to be  $3.2 \times 10^{11}$  molecules cm-3 s. This is equivalent to 2.4 days assuming  $1.5 \times 10^6$  molecules cm-3 ambient average OH.

**Reference**

Burkholder, J. B., Sander, S. P., Abbatt, J., Barker, J. R., Huie, R. E., Kolb, C. E., Kurylo, M. J., Orkin, V. L., Wilmouth, D. M. and Wine P. H. "Chemical Kinetics and Photochemical Data for Use in Atmospheric Studies, Evaluation No. 18," JPL Publication 15-10, Jet Propulsion Laboratory, Pasadena, 2015.

Lambe, A. T., Chhabra, P. S., Onasch, T. B., Brune, W. H., Hunter, J. F., Kroll, J. H., Cummings, M. J., Brogan, J. F., Parmar, Y., Worsnop, D. R., Kolb, C. E., and Davidovits, P.: Effect of oxidant concentration, exposure time, and seed particles on secondary organic aerosol chemical composition and yield, Atmos. Chem. Phys., 15, 3063-3075, 10.5194/acp-15-3063-2015, 2015." 120: "TI or TOC in the particles was obtained by subtracting the amount on the back filter from that on the front filter"

I am confused about the calculation. As you said that "The front filter of the double filter pack collected the particles, while the back filter placed downstream of the front filter was supposed to adsorb the same amount of volatile species as the front filter", may I think of it this way: particles in the front filter and volatile species in the back filter. Why the TI in the particle is not the amount on the front filter? Why it needs to subtracting the amount on the back filter?

**Re:** We are sorry to make this confusion. The front filter collected particles + adsorbed volatile species; the back filter adsorbed volatile species only.

We change the sentence in line 110-113.

"The front filter of the double filter pack collected the particles and also adsorbed some volatile species as positive artifact, while the back filter placed downstream of the front filter was supposed to adsorb the same amount of volatile species as the front filter."

128: "Only the compounds that existed solely in the front filter or with ion intensity in the front filter higher than that in the back filter by a factor of 3 were regarded as the organic compounds in the particle phase"

Please cite suitable literature.

Re: we cite the following paper

"Wang, X., Hayeck, N., Brüggemann, M., Yao, L., Chen, H., Zhang, C., ... Wang, L. (2017). Chemical characteristics of organic aerosols in shanghai: A study by ultrahigh-performance liquid chromatography coupled with Orbitrap mass spectrometry. Journal of Geophysical Research: Atmospheres, 122, 11,703–11,722."

**Results and discussion**

135, 138: "new particles larger than 14 nm were observed only 58 minutes after the injection of ozone flow", "With a prolonged residential time of 67 min..."

The authors talked about the results after 58 or 67 minutes. But the maximum of axis about the elapsed time in the Figure. 2 was 50.

**Re:** This is because Time zero of Figure 2 was set as the time when gaseous products first appeared, not when ozone was injected. We could not observe any new particle or gas-phase product during the first 48 minutes after  $O_3$  were added. So we did not show the data during that early period in Figure 2.

In line 142-147 we add:

"No particles formed in the absence of room light or  $O_3$ . Therefore, light was on throughout the experiments reported in the article. In the static mode experiments, we could not observe gas-phase products until 48 minutes after  $O_3$  injection. New particles larger than 14 nm were observed only 58 minutes after  $O_3$  injection. Afterwards, new particles begun to grow to form a typical banana-shape particle size spectrum (Figure 2a). This relatively long waiting time was likely due to the build-up of  $O_3$  and oxidation products. Time zero of Figure 2 was thus set as the time when gaseous products first appeared."

136: "No particles were formed in the absence of room light or ozone".

I don't see the relevant results (table or figure) shown in the paper.

**Re:** our pilot study showed that no particles formed in the absence of room light or  $O_3$ . We did not collect continuous time evolution data of particle and gaseous products in those pilot experiments. So we did not show those results.

On the other hand,  $O_3$  and room light were always supplied for the chemical measurement experiments shown in the manuscript,

154: "But those small new particles are expected to grow into CCN active sizes, given longer residence time and uptake of more condensing vapors in the atmosphere".

Please cite suitable literature.

**Re**: this paper is cited now:

"He, X. C et al. Role of iodine oxoacids in atmospheric aerosol nucleation, Science, 371, 589-595, https://doi.org/10.1126/science.abe0298, 2021."

156: "3.2 Macroalgal emission"

I think it is more suitable to remove this section to the first part of the Results and discussion

**Re:** Thank you for pointing out this. Now the order of subsections in "Results and discussion" is changed to

3. Results and discussion

3.1 Macroalgal emission

3.2 Gaseous products

3.2.1 Gaseous inorganic molecules and radicals

3.2.2 Gaseous organic products

3.3 Particulate products

3.3.1 Relative mass contribution of organic carbon and iodine to new particles

**3.3.2 Particulate organic products**

187, 188: "IO, IO2 and ClIO could be from the reactions between I, ClI and O3", "ClNO2 was likely to form upon similar reaction between Cl and NO2 in the bag reactor"

Give the reaction mechanisms or cite literatures.

Re: we revised the two paragraphs in line 181-193, following community comments

"2.  $IO_2$ , IO and CIIO in gas phase: these species showed a similar time evolution to I and Cl atoms. They could be from the reactions between I, CII and  $O_3$  (Saiz-Lopez et al., 2014). Sequential oxidation and aggregation reactions might have formed other halogen oxides (Gómez Mart n et al., 2013), but they might not be detectable by iodide-CIMS.

3. INO2, ClNO2 and INO3: INO2 and ClNO2 were detected in gas phase with similar time evolution with halogen atoms and halogen oxides (Figure 2d). INO3 was found in both gas and particle phases. INO2 and INO3 were usually thought to form upon the reactions  $I+NO_2+M \rightarrow IONO+M$  and  $IO+NO_2+M \rightarrow IONO_2+M$  in the atmosphere (Saiz-Lopez et al., 2012), which seems to be unlikely in our bag reactor because NO2 was not added. Considering NO3- was ubiquitous in the bag reactor of our experiment, it is likely that INO2 and INO3 formed via  $I_2O_2+NO_3^- \rightarrow IO_3^-+IONO$  and  $I_2O_3+NO_3^- \rightarrow IO_3^-+IONO_2$ . These reaction pathways have been supported by theoretical calculation and flow tube mass spectrometry experiments (G ómez Mart ń et al., 2022; G ómez Mart ń et al., 2020). ClNO2 was likely to form upon similar reaction between  $Cl_2O_2$  and  $NO_3^-$  in the bag reactor."

195: "which is contrary to the observation by HPLC-ICP-MS that total iodine was mostly dominated IO3- peak"

Could the authors explain the contrast?

Re: In line 199, we add

"We speculate that  $HIO_3$  might have been dehydrated to  $I_2O_5$  under thermal desorption temperature up to 180 °C in FIGAERO."

259: Scheme II

The formulas are too simple to understand the mechanism of particle formation. It might be meaningful to give formulas like Scheme I for several specific species.

**Re**: it is not appropriate to give exact reaction equations in *Scheme II* for the following reasons:

1. Particulate dimer products like  $C_{14}H_{16-26}O_{6-12}$  and  $C_{16}H_{20-32}O_{6-12}$  have higher carbon number and thus more complex molecular structure than those small molecules in *Scheme I*. CIMS analysis provided molecular formula information only. It is thus not realistic to propose an exact molecular structure and reaction equation. Any speculative molecular structure or reaction equation is unfounded and may be misleading.

- 2. For the highly complex system with tens and even hundreds of precursors, it is not realistic to present a general or uniform reaction equation to explain the complicated interactions among numerous precursors, intermediates and products, or to list all reaction formulas that occurred in the bag reactor.
- 3. A variety of accretion reactions without uniform oligomerization pattern (e.g., esterification, aldol condensation, hemiacetal reactions, peroxyhemiacetal formation and SCI reactions, etc.) might have occurred in the particles. It is again not realistic to present equations for just several specific species.

On the other hand, the two equations in our old manuscript, strictly speaking, can not be called a "scheme". In line 292-298, we rephrase to :

"As an example, we used two simplified reaction equations to illustrate addition-type cross-oligomerization between  $C_6$  and  $C_8$  monomers and self-oligomerization of  $C_8$  monomers, respectively:

 $C_{6}H_{6-12}O_{3-6} + C_{8}H_{10-16}O_{3-6} \rightarrow C_{14}H_{16-26}O_{6-12}$

 $C_8H_{10\text{-}16}O_{3\text{-}6} + C_8H_{10\text{-}16}O_{3\text{-}6} \xrightarrow{} C_{16}H_{20\text{-}32}O_{6\text{-}12},$

in which the  $C_6$ ,  $C_8$ ,  $C_{14}$  and  $C_{16}$  formulas are among the most abundant ones observed in the particle phase by the iodide-CIMS."